# Development and Application of a New Apparatus for Moisture Measurement in Building Composites

**Miroslav Frydrych \*, Miroslav Herclík, Michal Klaban, Roman Knížek and Ludmila Fridrichová**

Department of Textile Evaluation, Faculty of Textile Engineering, Technical University of Liberec, Studentska 2, 461 17 Liberec, Czech Republic; miroslav.herclik@tul.cz (M.H.); michal.klaban@tul.cz (M.K.); roman.knizek@tul.cz (R.K.); ludmila.fridrichova@tul.cz (L.F.)

\* Correspondence: mirolav.frydrych@tul.cz; Tel.: +420-725-843-077

**Abstract:** The presented paper is divided into two levels. On the first level, the research is focused on development of a material that includes a smart fabric membrane (nanofiber) and is waterproof and vapour-permeable. On the second level, it deals with the development of an apparatus which is able to evaluate the resistance of the building composite against water. A new device is presented that can measure how waterproof the building material is, with a thickness between five to 20 centimetres. The tested samples are in the shape of a square or circle, they have a width of 20 centimetres, or a diameter of 20 centimetres. All of the building composite samples were tested using a water column that is one meter high. Experiments simulate floods on this new apparatus. It was found that materials can be evaluated not only according to the amount of water flow per unit of time, but also according to the shape of the flow curves.

**Keywords:** nanofibrous membrane; water flow; hydrostatic resistance; composite material

## 1. Introduction

Moisture in a building structure accelerates its ageing and degradation [1]. Moisture is one of the main decay factors in building envelopes, increasing the repair costs of structures to several billions. Such damage is apparent in historical and traditional buildings, since they are exposed to the moisture influence for longer time periods, as well as in contemporary building envelopes [2]. Composites with ultrafine polymer fibres may perform interesting properties, which could be an advantage in some applications [3,4]. Functional clothing, for example GORE-TEX membrane, is known throughout the world. The garment was made of a fabric that has the ability to retain moisture and also release water vapour. [5]. The clothes tag thus added another figure, i.e., the height of the water column that the fabric holds. This was the main idea for the project—to use the principle of textile membrane as a protection against dampness of the building. Nanofibers are a very promising type of material with a broad range of possible applications. [6]. Generally speaking, the permeability of membrane must depend not only on the water content, i.e., the degree of hydration, but also on the geometrical structure of the membrane and subsequent conditions [7].

In the building industry, we encounter materials that are water repellent, but they are presented in such a way that creates the feeling of materials being waterproof. These are mostly materials applied by spraying, i.e., they work only temporarily, not permanently. The methodology and results of the first experiment were presented for building materials with a nanofiber membrane [8]. Two types of composites were investigated. The gravimetric method was used for evaluation of water permeability in the material. The amount of water that penetrates the building composite over a defined area over a given time (up to six days) under hydrostatic pressure (P = 14750 Pa) was also determined. The obtained results showed that a newly developed "breathing" building composite

with nanofiber membrane can be used as a barrier against moisture contamination in masonry with long-lasting effects of water.

The durability of concrete depends to a large extent on the difficulty of external aggressive agents in penetrating the porous network of concrete. The ability of this material to withstand the intrusion of these aggressive agents is mainly characterized by water permeability. [9].

In the subsequent study, geopolymer samples were used. This material has the same property as concrete, as it is porous. When the nanofiber membrane was used, the waterproof material obtained was able to withstand the long-term effects of water [10].

There are a number of instruments for measuring the resistance of materials to water column pressure. Such instrument include the KJ -3093 Hydrostatic Head Tester from Kejian, and the Automatic Hydrostatic Head Tester from TEXTEST AG, which can detect penetrated water drops. Another possible device is the Hydrostatic Head Tester M018 from SDlAtlas.

Wang et al. [11] described the water permeability measurement method through cracked concrete, and they claim that the higher permeability of concrete is indicative of its poor quality due to the presence of additional pores, voids, and cracks. Lepech and Li [12] tested the water permeability of concrete samples on a similar apparatus. However, they selected a reservoir with a constant level of water for high throughput samples. For samples that showed low permeability, they used the decreasing water column method. Palin et al. [13] presented a rapid test for establishing the water permeability of cracked mortar specimens. The samples were placed in a section of the pipeline and covered with silicone, which was allowed to cure for 24 h. The water flowing through the sample flowed into the collecting containers and its weight was recorded after five minutes. The water level of each column was maintained between 1.05 and 1 m, giving an almost constant water head of 0.1 bar. Another interesting method of testing is described by Tziviloglou et al. [14]. This is a measurement of the water permeability of cracked concrete. Prior to the measurement, the tested prisms with dimensions of 40 × 40 × 160 mm were deformed by the so-called three-point bend to create defined cracks. The water flowed through the test sample into a collection vessel placed on an electronic scale, which recorded the weight gain of the water that flowed over time using a simple program. There are, of course, many other methods of measuring the permeability of materials that are based on a different basis, as shown by Cao et al. [15]; Mechtcherine et al. [16]; Amriou et al. [9]; and Phung et al. [17]. A useful method for how to measure effective porosity and the permeability coefficient of pervious concrete was described in Tan et al. [18].

Parameters of the different devices in the different experiments were reported as follows: the permeability was determined experimentally using a constant load permeameter. Cylindrical specimens (L = 100 mm and radius = 50 mm) were placed in a cylindrical mould, which is subjected to constant water pressure. The flow Q was obtained from the water mass collected as a function of time [19]. The tests are carried out on samples that are 16 cm in diameter and 14 cm in height with a longitudinal opening of 6.3 cm in diameter [9]. The sample mounting plate between the two sub-chambers was made of 6.4 mm thick perspex, which contained six 64.5 mm diameter samples [20]. As shown, the usual sizes of the tested samples are from 5 to 16 cm and the thickness is a maximum 15 cm. On the apparatus, it is able to test four square or circle shaped samples 25 centimetres in size and 20 centimetres in thickness. Galbraith et al. [20] have presented a methodology for carrying out moisture permeability measurements under a range of barometric pressures. The technique has been successfully applied to a particle board test material. Current methods for assessing the permeability of cracked concrete generally have long testing periods and poor experimental reproducibility. The authors [13] present a rapid test for establishing the water permeability of cracked mortar specimens. The principle of the method is based on the height of the water column.

There are many existing permeability measurement methods, but each method is only suitable for specific materials. The new measuring device has been designed to test building composites up to 20 cm thick and is fully automated. The tested materials will be evaluated according to the water flow per unit of time and the shape of the flow curves.

## 2. Materials

For the experiment, three layers of laminate as the water barrier were used, as is shown in Figure 1. Two layers were nonwoven and a nanofiber membrane was inside. All of the layers were connected by lamination. It was already presented [21] that for most multi-layer materials, the lamination process deteriorates some properties of the whole laminate, particularly decreasing the evaporation. However, the laminate that was used for the experiment had sufficient properties for the chosen purpose, i.e., the building composite.

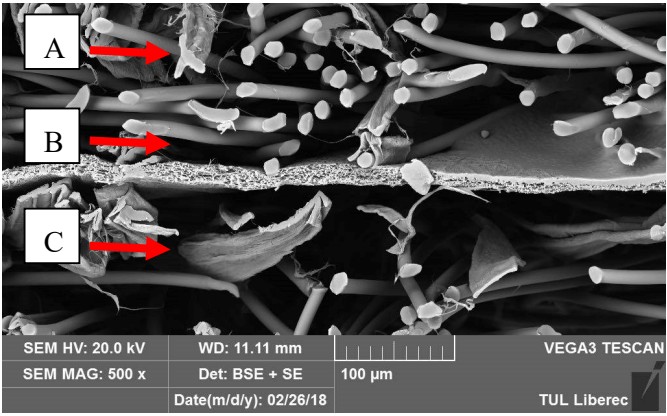

**Figure 1.** The water barrier consists of three laminate layers, i.e., nonwoven layer (**A**), nanofiber membrane (**B**), and nonwoven layer (**C**).

The measurement of permeability using the method of measuring the resistance of materials to the water column pressure was performed on the permeable water-resistant heat insulation panel Hýsek et al. [10], which could be applied in diffuse open buildings, where the vapor permeability of the nanofiber membrane will be utilized while transporting moisture from the building to the external environment. Fridrichová et al. [8] presented that the hydrostatic resistance of a membrane could be used to protect a building against water, for example during floods.

A nanofibrous membrane from NANOMEMBRANE s.r.o. was used for testing. The use of nanofiber membranes is also suitable for wooden constructions and therefore, commercially sold OSB boards (class 3, density 620 kg/m³) were also used for comparison with these nanofibrous membrane. The nanofiber membrane was made of polyurethane via electrospinning using Nanospider technology by Jirsák et al. [22]. The polymer solution and spunbond technology was used. The electric field with a voltage of 80.7 kV, and the distance of the condenser of 190 mm was used.

## 3. Methods

The developed measuring system is able to correctly evaluate the water resistance of the building composite. It is known that there are many measuring systems for evaluating the water permeability of building materials, see below. On the first level, a different size and shape of the tested samples is used. On the second level, a device measures the long-term impact of the water column (1 to 5 m high) on the building composite, i.e., samples are subjected to constant water pressure, for example for six days.

### 3.1. Description of the Measuring Assembly

A functional control system for measuring water permeability of four samples of building composite was designed. As shown in Figure 2, the new device enables measurement of the actual amount of liquid that the test sample passes over time at a defined hydrostatic pressure. The present apparatus is designed to measure samples with a maximum thickness of 20 cm, and for different sample shapes (square, rectangle, or circle).

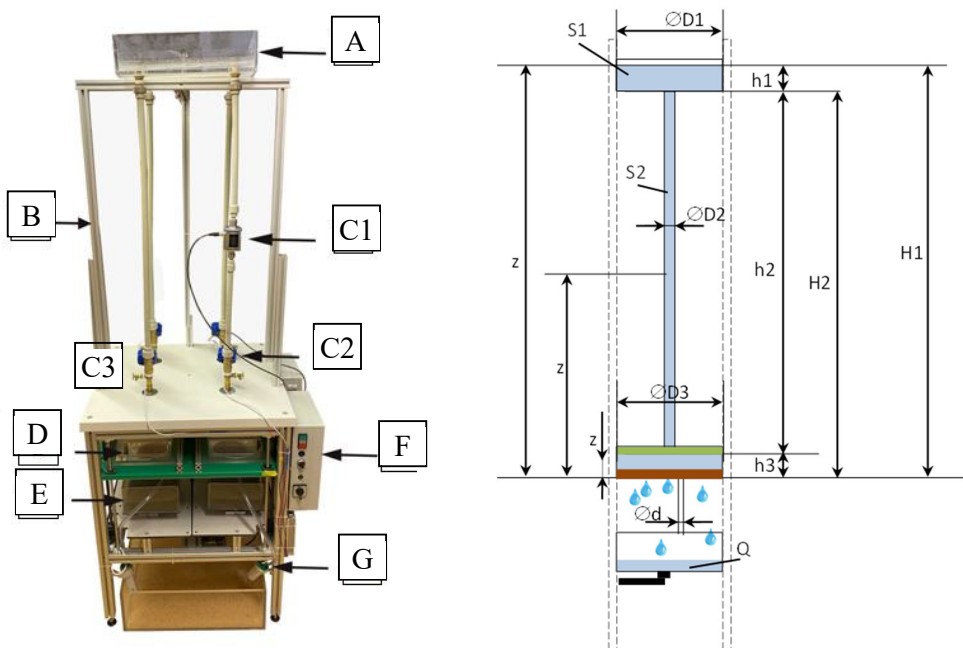

**Figure 2.** (**A**) HP4 apparatus, (**B**) scheme of hydrostatic pressure on sample.

As illustrated in Figure 2A, the carrier part is made of light alloy and the horizontal parts are made of plastic boards. The device has nine plastic containers. The highest point of the device is the liquid container (A), which is a common container for four measuring sets (D). Each of the four sets consists of three parts: hydrostatic, testing, and measuring.

The hydrostatic part, consisting of a top container (A) water reservoir, pipes (B) and two types of valves (C2, C3) is connected by screwing with a rubber gasket. Since the liquid container is located on the sliding profiles, it is possible to adjust the additional pipe (B) and thereby change the height of the water column or the hydrostatic pressure applied to the test samples. Two valves are located above the plate. The plastic board forms the main support of the device, which is also a dividing plane between the hydrostatic and the testing part of the instrument.

The upper larger valve (C2) is used to close the pipeline, and it is therefore not necessary to drain all of the water when changing the sample, or some of the assembly can be shut off from the measurement process. The smaller lower valve (C3) serves to vent the assembly during water filling. The valve (C1) is a flow meter that records the water absorption of the material.

The testing part consists of four containers for the test samples (D), which are clamped between two horizontal plates. The plates are controlled by a mechanism by an electric motor. The top horizontal plate has a 3 mm silicone layer in addition to the supporting profiles on the underside. This provides a watertight layer between horizontal plate and the upper edge of the test containers. The silicone layer is fully laid across the underside of the board, allowing variability in the use of different shapes of test containers (circle, square, rectangle).

The measuring and evaluating part consist of collection of four containers for water and platforms with a weighing sensor (E). The weighing sensor records the mass increments of the liquid Q [g] at time t [s]. The weight gain data is stored in the control unit (F) for further processing. The output is a graph describing the dependence of weight of the flow-through liquid on time, which can be observed on-line on the display.

The electronics also control the height of the liquid in the upper reservoir. This container is equipped with a float that sends a signal to the control unit, then the change in the level is evaluated, and in the event of its drop, the pump starts, and the water is added from the bottom lower reservoir (G).

### 3.2. Computation Model

Król et al. [23] presented a method for determining the vertical and horizontal flow of water through the specimen of porous material, permeability test was used following the EN 12697-19 standard. In this method, a column of water with a constant height (300 mm) is applied to the specimen for a controlled time (the range from 30 to 60 s) in either the vertical or horizontal direction, depending on the parameter being measured.

The quantity of water flow in the vertical and horizontal directions, expressed as QV and QH, were determined and calculated from:

$$Q_{(V,H)} = \frac{m}{t} \times 10^{-6},$$  (1)

where m is the mass of the flowed water, in g; t is the time of water collection, in sec.

The apparatus is able to measure the actual amount of liquid flow through the building composite; the principle is similar to the device presented above. In order to verify the accuracy of the results obtained on the prototype, a simulation model was created based on the calculation of the time outflow of liquid Q [g] from a system of three circular cross-section containers, where their diameter ($\varnothing$ D1, $\varnothing$ D2, $\varnothing$ D3), height (h1, h2, h3) and cross-sectional area (S1, S2, S3) are defined. The total height H1 defines the initial hydrostatic pressure in the system, see Figure 2B. The size of the flow openings $\varnothing$ d and their number n are defined in the drain container.

It is considered that the case when the entire assembly (Figure 2B) is filled with liquid up to the upper container, i.e., up to the height H1; after filling, the assembly is without any further inflow ($q0 = 0$) of the liquid.

For the i-area of the circular cross-section $S_i = 0{,}25.\pi.D_i^2$; $I = 1,2,3$; $s = 0{,}25.\pi.d^2$, the height limits, see Figure 2B, are $H1 = h1 + h2 + h3$, $H2 = h2 + h3$, $H3 = h3$.

Depending on the level of z, during the discharge of liquid from the system of three containers, the area of their cross-section S (z) changes according to:

$$\begin{aligned} S(z) &= S_1 \text{ pro } H2 \leq z \leq H1, \\ S(z) &= S_2 \text{ pro } H3 \leq z \leq H2, \\ S(z) &= S_3 \text{ pro } \quad 0 \leq z \leq H3. \end{aligned}$$  (2)

The time change of the water level is based on the differential equation

$$\dot{z}(t) = \frac{q_0}{\rho.S(z)} - \frac{s.n}{S(z)}\sqrt{2.g.z}\,, \qquad z(0) = H1.$$  (3)

The time course of the weight of the leaked liquid is similarly based on the differential equation

$$\dot{Q}(t,z) = \rho.s.n\sqrt{2.g.z}\,, \qquad\qquad Q(0) = 0.$$  (4)

In the case where a constant mass inflow $q_0 > 0$ flows into the upper container, Equation (2) for z (t) = 0 can express the steady level of water

$$\bar{z} = \frac{1}{2.g}\left(\frac{q_0}{\rho.s.n}\right)^2.$$  (5)

The functionality of the measuring apparatus through an above-mentioned simulation (computation) model was further verified. This simplified porosity model was used for the planned experiment to exclude various measurement effects, such as swelling of material or seals around the circumference of the test sample. That is why the test containers with defined holes diameters (1 and 5 mm) and a specified number of holes (1 to 9) were created. The results of the experiment and simulation model are show in Figure 3.

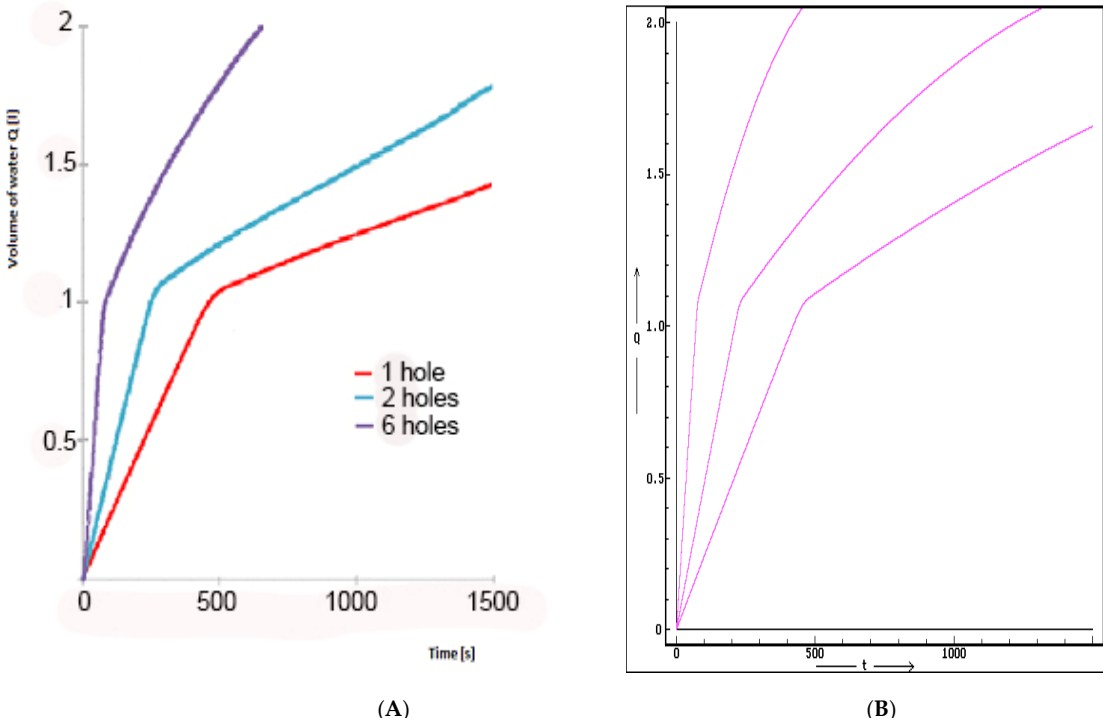

**Figure 3.** Water flow through the test vessel through a variety of holes. (**A**) Experimental, (**B**) computational.

As is shown in Figure 3, the experimental results obtained were compared to the calculation model. Sealing effectiveness is one of the critical factors in the permeability measurement. On this computation simulation, it was able to test various seals used to fix the test sample in the container.

## 4. Results and Discussion

In the first experiments, only the effect of the barrier was monitored. A six-aperture test container was used with a hole diameter of 1 mm. Different barriers (cloth linen fabric, non-woven fabric with hydrophobic finish, nanofiber membrane, vapor permeable building foil) were gradually introduced into the container. The results are shown in Figure 4.

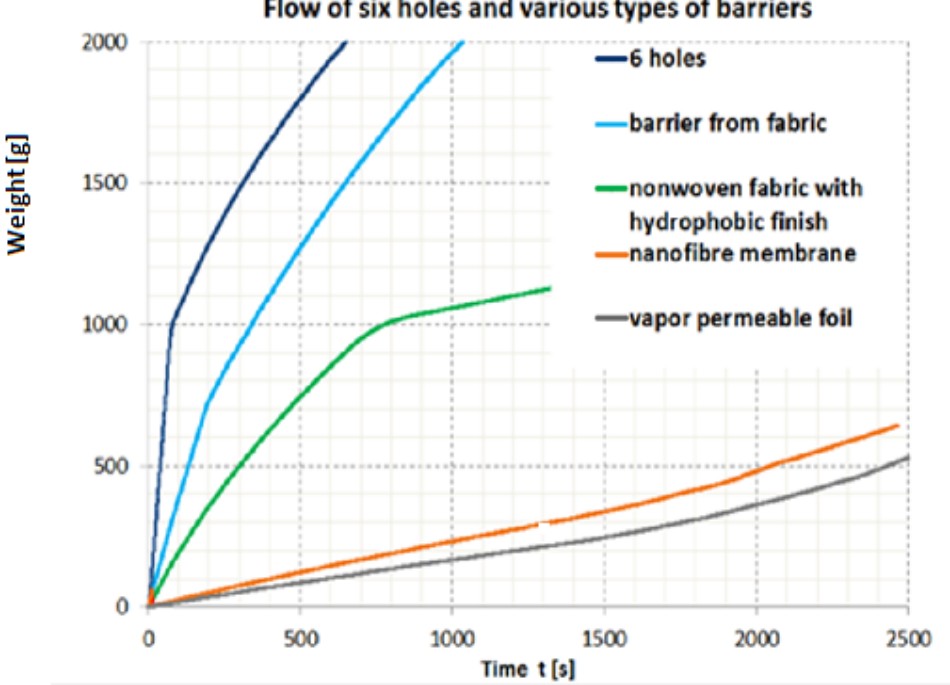

**Figure 4.** Flow rate through a test container with six openings and different types of barriers.

The measuring system was filled with water to height H1, shown in Figure 2B. The flow of water through the building composite was monitored and the weight and flow time of water were recorded. As can be seen from Figure 5, 1000 g of water in 90 sec ran out from a six-hole container with a diameter of 1 mm. This means that all of the water has leaked from the upper container. Consequently, the height of the water column decreases significantly. This phenomenon confirms the break in the graph curve, as shown in Figure 4.

When using the fabric as a hydro-barrier, 1000 g water per 330 s, i.e., at triple time with respect to the outflow of water from the container without a barrier, flowed out. When a non-woven fabric with a hydrophobic finish as a barrier was used, 1000 g of water leaked per 700 s, which is 10-fold time. The best results were achieved with the nanofiber membrane and building foil. The discharge time of the liquid was about 2300 s, i.e., over 30 min.

### 4.1. Oriented Strand Boards (OSB)

Commercially sold OSB boards from Kronospan Jihlava (class 3, density 620 kg/m3) were tested and, for comparison, the boards covered with the nanofiber membrane from NANOMEMBRANE s.r.o were tested. Testing was performed at a water column height of 1 m and a test sample area of 154 cm². The height of the water column was constant during the measurement. After 24 h, only 46 g of water flowed through the nanofiber membrane test sample (Figure 5). When testing samples without hydro barriers, testing is terminated when 4000 g of water flows through the test sample. For OSB without a membrane, this amount flowed in only 12,600 s. (Figure 6). Testing was carried out at a constant height of 1 m water column and a sample area of 154 cm².

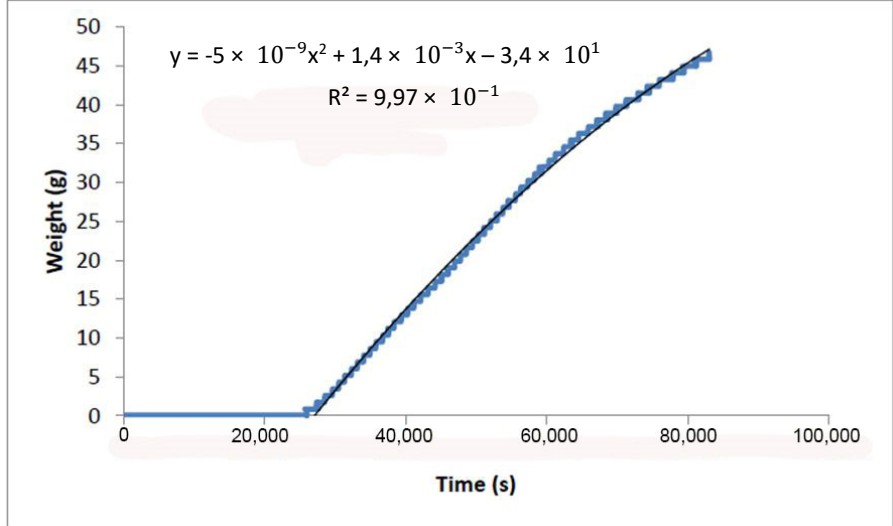

$$y = -5 \times 10^{-9}x^2 + 1,4 \times 10^{-3}x - 3,4 \times 10^1$$

$$R^2 = 9,97 \times 10^{-1}$$

**Figure 5.** Water flow through OSB with nanofibrous membrane.

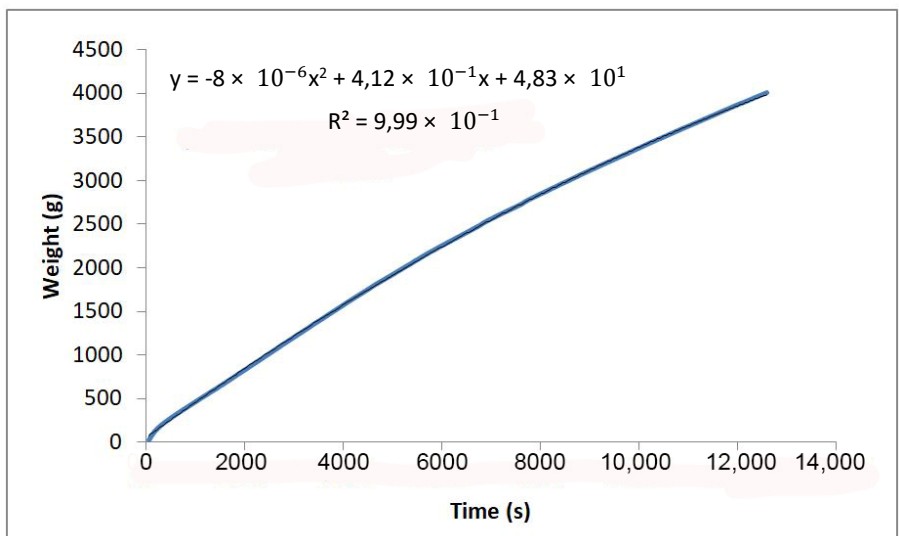

$$y = -8 \times 10^{-6}x^2 + 4,12 \times 10^{-1}x + 4,83 \times 10^1$$

$$R^2 = 9,99 \times 10^{-1}$$

**Figure 6.** Water flow through untreated OSB.

*4.2. Permeable Water-Resistant Heat Insulation Panel*

The amount of 486 g of water flowed through the panel in 24 h. The resistance of the sandwich composite to the water column is due to the nanofiber membrane used, as demonstrated by testing the same panel with the missing membrane layer (Figure 7). When the sandwich without the nanofiber membrane was loaded with a water column, the amount of 4000 g flowed in 260 s. Testing was carried out at a constant height of 1 m water column and a sample area of 154 cm².

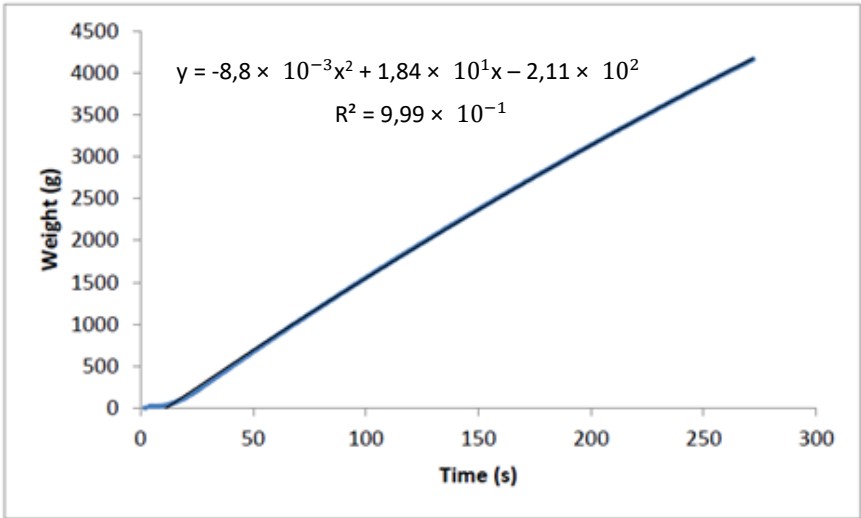

**Figure 7.** Water flow through permeable water-resistant heat insulation panel with missing membrane layer.

The so-called "wick" parts of the curve were observed in the graphs of samples with nanofibrous membrane. These are parts of the zero-weight curves in the collecting containers, but some water flow was recorded on the flowmeter in front of the test sample. This is the length of time the composite has soaked in, followed by an overflow that has already been recorded by the weight sensor. Each material pulls water into its structure with a different force, which is reflected in the flow curves. The amount of water flowing at the end of the measurement is dependent on the time of the wick effect when the first flow occurs. The different composition of the tested materials has not only influenced the total amount of water flowed, but also the shape of the flow curves and different "wick" parts. When testing the water column resistance of the nanofiber membrane itself, it was found that no water flows through the used nanofiber membrane. When using a nanofiber membrane in a sandwich, however, water flows through the membrane. This phenomenon can be explained by the deformation and enlargement of the pores in the membrane due to its application to the rough surface. According to the performed expectations, the underlying material, which is hygroscopic and actively absorbs water, also plays a role. These forces can help transport water through the membrane.

### 4.3. Outlook to Envisaged Device Development

Further development of the sample sealing system is expected in the future. At present, the samples are sealed in the mould with silicone, but there are some waxes that allow for a better waterproof seal. Sealing waxes are the direction where an attention will turn in the future.

The height of the water column is currently adjustable according to the length of the pipes that are implemented in the machine, and in reality, a few variants of water column heights are possible. In the future, it is planned to equip the device with a pipe, where it will be possible to continuously change the height of the water column.

Another challenge is the development of software for device control and recording of measured data. A software on a universal free platform is currently used, which is fully sufficient for performing experiments, however, it is planned to develop a software with a better user interface.

### 5. Conclusions

A completely automated apparatus for measurements of moisture in building composites was developed. The functionality of the instrument was tested experimentally, and several materials were characterized by this new method. It was shown that the dependence of the amount of water

flowing on time has the course of the second-degree polynomial. This model describes the observed phenomenon with a very high degree of tightness (determination coefficient 0.997 or more). In these experiments, a so-called wick effect was also identified, which causes water flow through the nanofiber membrane. The wick effect can be quantified over time and is characteristic of each material. The developed apparatus for moisture measurement in building composites is endowed with an interface that enables the user to set up a behaviour of the system, for example the refilling of water to secure the constant water column, drainage of the water from containers above the weight sensors after reaching certain weight, or turning on of the pumps by the user. The functionality of the device was verified by measuring oriented strand boards panels with and without nanofibrous membrane.

**Author Contributions:** Conceptualization, L.F.; data curation, M.F. and L.F.; investigation, M.F. and M.H.; methodology, M.F., M.H., R.K. and L.F.; software, M.K.; writing—original draft, Miroslav Frydrych; writing—review and editing, R.K. and L.F. All authors have read and agreed to the published version of the manuscript.

**Funding:** This research was funded by funds provided by the Technical University of Liberec—SGS project No. 21250 and SGS project No. 21302.

**Acknowledgments:** This work was supported by the Ministry of Education of the Czech Republic within the SGS project No. 21250 and SGS project No. 21302 at the Technical University of Liberec.

**Conflicts of Interest:** The authors declare no conflict of interest.

## References

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
