# Peer review of "Development and Application of a New Apparatus for Moisture Measurement in Building Composites"

_applsci, doi:10.3390/app10155288_

Round 1
Reviewer 1 Report
Reviewer appreciates the authors' effort on nano-fiber membrane application to protect against building dampness. Following are some suggestions to improve the quality of paper.
- Page 3 of 14, line 111, please rewrite the sentence "...during floods Fridrichova et al. [4]."
- Page 4 of 14, on paragraph started at line 124, citations are provided which helped to obtain experimental guidelines. Rewriting is required as "This paper [20] has presented..." "The authors [9] present.." are not a good approach to start a sentence considering that this is not a literature review/introduction section.
- Page 5 of 14, line 148, "Figure 1" should be "Figure 2"?
- Why you start with new Section/subsection on "Theoretical Part"? Please reorganize the paper.
- Page 7, line 210, please explain more about which methodologies were used in "simulation (computation) model?
- Page 8, line 218 and line 219, "the results are very good". isn't in a vague statement? line 219: "Thanks to this computation simulation". Is not a good way to present. Please rewrite.
- Conclusion are not well written. Thanks!
Author Response
Dear reviewer,
many thanks for your valuable comments. We incorporated your suggestion to the manuscript. Please find attached response to your comments.
Best regards
Miroslav Frydrych

Reviewer 2 Report
See attached document

Author Response

(The authors gave the same response as above.)

Reviewer 3 Report
Please find the attachment.

Author Response

(The authors gave the same response as above.)

Round 2
Reviewer 1 Report
Thanks for addressing all the comments.
Author Response
Thank you for your review, the manuscript was spell checked.
Reviewer 3 Report
Please find the attachment.

Author Response
Dear reviewer, thank you for your comments. Please see attached response to your comments.
